# Assessment of Fungal Succession in Decomposing Swine Carcasses (*Sus scrofa* L.) Using DNA Metabarcoding

**DOI:** 10.3390/jof9090866

**Published:** 2023-08-22

**Authors:** M. Denise Gemmellaro, Nicholas Steven Lorusso, Rachel Domke, Kristina M. Kovalska, Ayesha Hashim, Maria Arevalo Mojica, Amanda Joy O’Connor, Urvi Patel, Olivia Pate, Gloria Raise, Maria Shumskaya

**Affiliations:** 1Department of Biology, Kean University, Union, NJ 07083, USA; 2Department of Natural Sciences, University of North Texas at Dallas, Dallas, TX 75241, USA

**Keywords:** forensic mycology, animal decay, carrion mycobiome, necrobiome metabarcoding

## Abstract

The decomposition of animal bodies is a process defined by specific stages, described by the state of the body and participation of certain guilds of invertebrates and microorganisms. While the participation of invertebrates in decomposing is well-studied and actively used in crime scene investigations, information on bacteria and fungi from the scene is rarely collected or used in the identification of important factors such as estimated time of death. Modern molecular techniques such as DNA metabarcoding allow the identification and quantification of the composition of microbial communities. In this study, we used DNA metabarcoding to monitor fungal succession during the decomposition of juvenile pigs in grasslands of New Jersey, USA. Our findings show that decomposition stages differ in a diversity of fungal communities. In particular, we noted increased fungal species richness in the more advanced stages of decomposition (e.g., bloat and decay stages), with unique fungal taxa becoming active with the progression of decay. Overall, our findings improve knowledge of how fungi contribute to forensically relevant decomposition and could help with the assessment of crime scenes.

## 1. Introduction

### 1.1. Decomposition in Forensic Investigations

The process of decomposition is a complex series of biochemical reactions that contribute to ecological systems through the recycling of nutrients. Decomposition is facilitated by a number of saprobic taxa, such as bacteria, invertebrates and fungi [1,2]. Fungi are major decomposers of organic plant material, with species succeeding each other during the decomposition stages [3]. Some species of fungi inhabit decaying trees that have only previously been decayed by certain other species. The successor emerges only after the preceding fungus has died [3]. The gradual change in the diversity and composition of fungal guilds during wood decay has been intensively studied; the decomposition of the wood consists of several stages with different complex polysaccharides oxidized by fungal communities which possess the necessary enzymes [4,5,6,7], though the contribution to animal decomposition is less well understood.

Animal tissue is rich in proteins, and its decomposition is mediated mostly by bacteria and invertebrates; however, certain fungal species might participate in the process as they could be attracted by simple sugars exposed at different stages of decay. The decomposition of human corpses or other mammals is a complex process that is described in forensic sciences by a predictable sequence of stages. The succession of these stages can be used to build a decomposition timeline, which, in turn, can be used in forensic death investigations to estimate the time of death [8,9]. The stage immediately after death is defined as the fresh stage, and it usually displays no obvious putrefactive phenomenon [8,10]. The second stage is the bloat stage, caused by the anaerobic activity of bacteria which produce gases inside the body. These gases exert great pressure which eventually causes the skin to break. This marks the start of the decay stage; during this stage, gases and fluids are purged from the body, and the body deflates, but soft tissue is still present. At this point, putrefaction becomes the driving force of decay. Putrefaction is a microbial process through which proteins, carbohydrates and lipids are fermented into simpler compounds; this process is started by microorganisms living in the intestine, which then migrate to the lymphatic system and reach other tissues [11]. This process results in an increased amount of biologically available macromolecules that can be exploited by opportunistic species such as fungi. Once soft tissues are consumed by putrefaction, the stage of advanced decay begins and continues until all or the majority of soft tissues have been consumed and bones are entirely or partially exposed [12]. This last stage is commonly defined as skeletonization or the “dry” stage [8]. The stages of decay occur in a predictable order, such that all corpses decaying in similar conditions proceed through the same progression of decomposition [13].

During each stage, a different ephemeral ecosystem is created on the corpse, and it has been observed that microbial communities on the body vary throughout the progression of decomposition [14,15]. The study of microbial communities that facilitate the decomposition of a human body could potentially offer support to forensic investigations. However, decomposition studies using human cadavers are limited to a few places in the world where human taphonomic facilities (HTF) are allowed [16,17,18]. Where HTFs are not allowed, research focuses on animal decomposition. Work by Payne was the first to use the animal model as a substitute for humans, and his results demonstrate that the results obtained from non-human vertebrates are indeed applicable to humans [19,20]. In fact, even though vertebrates can have different morphological features depending on the group they belong to (e.g., long fur, horns, antlers), all vertebrate carcasses possess soft tissue and bones, and this makes it possible to build decomposition models using one taxon and potentially apply these to different taxa [21,22].

Decomposition is affected by a combination of both abiotic and biotic factors; among the abiotic factors, temperature and weather conditions are of particular importance [23]. Depending on the temperature, time of the year and humidity, decay stages can last variable durations, from several days to several months. Abiotic factors affect biotic drivers of decomposition, with insects being a primary taxonomic group that has been extensively researched in relation to their role in decomposition ecology [24]. Different groups of insects (e.g., blow flies (*Calliphoridae*) and families including carrion beetles (*Silphidae*)) appear in a predictable order through the decomposition process, and some will lay eggs in (or colonize) the remains; insects developing on the corpse do so at predictable rates that depend on the insect species, temperature and other environmental factors [25,26,27,28,29,30,31]. Immature fly stages, such as maggots, contribute meaningfully to cell and tissue degeneration and contribute to the exposure of bioavailable nutrients for other decomposing taxa. Both the ecological succession of insects and other arthropods and the rate of insect development can be used to estimate the time of colonization (the TOC or time since the eggs are laid on the body) or the minimum post-mortem interval (mPMI) [32]. Insect activity is one of the major driving forces behind the process of decomposition; studies have shown that when insects do not have access to a corpse, it decomposes at a lower rate compared to a corpse that is accessible [33]; however, even in the absence of insects, decomposition does continue, highlighting the supporting role of microscopic decomposing taxa [34]. It is likely that insects also contribute to decomposition by redistributing microorganisms around the corpse and/or across the different carcasses they visit. Given the utility of using this well studied group, expansion to other previously understudied decomposing taxa is required.

Despite their important saprobic function and active contribution to the decomposition of vertebrate carcasses, microorganisms have received less attention in the literature [35]; in part, this is due to the historic intractability of including them in forensic analysis. This gap in knowledge makes it critical to explore as many factors contributing to the process of decomposition as possible to better understand decay and to offer better support to forensic investigations.

### 1.2. Fungi in Animal Decomposition

Fungi are a diverse kingdom of heterotrophic eukaryotic organisms with more than 3.2 million species currently estimated, including microscopic unicellular and filamentous species [36,37]. The contribution of fungi to decomposition has been shown to be significant to forensic investigations [38,39,40,41]. Due to their specific geographical distribution and the association of certain fungi with decomposing remains, fungi have been used in forensic investigations to establish connections between locations and individuals; this has been shown in several police reports mentioned by Hawksworth and Wiltshire [38]. For example, Vandevoorde and Vandijck were able to determine the time of death by examining the growth rate of fungi recovered on the body of a woman found indoors at a constant temperature [42]. However, in the presence of multiple fungal species growing at different rates, or in scenarios where temperature or other biotic and abiotic variables are at play, or even in cases where cryptic species contribute to decomposition, a simple approach cannot be effective [43]. While visible fungal bodies can be identified morphologically to support the post-mortem interval (PMI) identification [44], most of the fungi are microscopic, being present in single cells or hyphae filaments.

### 1.3. DNA Metabarcoding of Fungi

The challenge of identifying cryptic fungi can be overcome with the utilization of modern molecular biology techniques. The quantitative and qualitative composition of microscopic organisms from the environment, such as filamentous fungi, can be effectively assessed using a DNA metabarcoding method. In DNA metabarcoding, several taxa in one sample can be identified. Specifically, DNA from the microbial community is isolated, an internal genomic sequence (barcode) is amplified using a polymerase chain reaction (PCR), and this is then sequenced using high-throughput next-generation sequencing (NGS) and analyzed with subsequent bioinformatical tools, which are used to sort the DNA barcodes and compare them to a reference database such as GenBank (https://www.ncbi.nlm.nih.gov/genbank/), or UNITE database [45] for species identification. This method provides information on complex microbiomes [46,47,48,49,50] and allows for estimates of the richness and diversity of microbial communities.

For fungi, the ITS2 region is a genomic sequence that serves as a DNA barcode [51,52]. ITS2 has already shown promise in its utility for a metabarcoding-based estimation of PMI due to the difference in taxon richness and the relative abundance patterns of fungal communities during decomposition progression [53]. In addition, the diversity of fungal communities revealed by the metabarcoding approach has allowed researchers to discriminate the sites of cadaver decomposition where the cadaver was placed or from where it was moved [54]. The forensic science community would benefit from standardized, specific guidelines for metabarcoding data in legal disputations with regard to environmental conditions and geographic location [48], as the dynamics of fungal succession during the decomposition of animal bodies are still not fully understood [55]. While work on this area shows great promise, little is currently known about geographic variation on broad spatial scales (e.g., inter-continental differences) in how fungal communities assemble or what major taxonomic groups differ across space and time. This makes a standard method, employed across regions and ecosystem types, crucial for spotting and driving themes in fungal communities contributing to decomposition.

Here, we explore the successional patterns of fungi throughout the process of the decomposition of swine (*Sus scrofa* L.) carcasses in the Northeastern United States. We allowed swine carcasses, protected from vertebrate carrion feeders, to decompose in the natural grasslands of New Jersey to evaluate the following: (1) the biodiversity of fungi participating in the decay of animal bodies in New Jersey, (2) the hypothesis that fungal taxa change significantly across the major stages of decomposition; (3) the hypothesis that the distribution of fungi throughout the body is not significant during the process of decomposition. Beyond providing us with a chance to determine how fungal taxa change across these stages, this work also adds to a growing series of studies and provides regional-level data for forensically relevant fungal taxa at a regional scale.

## 2. Materials and Methods

### 2.1. Field Experiment Design

The experiment was conducted in a suburban area of Union County, NJ, USA, during the months of June and July of 2021. Daily temperatures varied from 13.8 °C to 39 °C in June and from 17 °C to 36 °C in July (Newark Airport Weather Station, 8 km from the research site). Twelve juvenile pig carcasses frozen at −20 °C were purchased from the University of Pennsylvania swine unit, Philadelphia, PA, USA. Once a corpse was taken out of the freezer and completely thawed (4–6.5 h), a “time zero” swab was collected as described below, and carcasses were left in an open field until complete decomposition was achieved. The carcasses were protected by metal cages to avoid predation by vertebrate predators while still allowing access to arthropods. The carcasses were checked twice daily to assess the decomposition state, and temperatures were recorded. At the beginning of each decomposition stage (five stages: fresh, bloat, active decay, decay, skeletal), three distinct body districts (mouth, abdomen, genitalia) (Figure 1) were sampled for fungal community analysis using a cotton swab of the surface. The complete decay of a corpse took between 6 and 8 days. The samples were stored in a −80 °C freezer prior to analysis. Four randomly chosen pigs were analyzed for fungal communities.

### 2.2. DNA Extraction

DNA swabs were collected using a sterile cotton swab from the surface of decaying pigs at each stage and location. The swabs were then stored in individual 15 mL Falcon tubes at −80 °C. In total, 75 swabs were processed. DNA was isolated using a PowerSoil DNA isolation kit (Qiagen, MD, USA) according to the manufacturer’s instructions. Swabs were cut from the wooden stems and mixed properly with the initial DNA isolation solution from the kit using BeadBug homogenizer (Benchmark Scientific, NJ, USA). The concentration of extracted DNA was measured via absorbance at 260 nm using NanoDrop spectrophotometer (ThermoFisher Scientific, MA, USA). Extracted DNA was stored in a 10 mM Tris buffer with a pH of 8.0 at −80 °C.

### 2.3. PCR

The PCR for the ITS2 gene region from the extracted DNA was carried out as in [49,56]. The primers were always fITS7 as forward (F), 5′-GTGARTCATCGAATCTTTG, and ITS4 as reverse (R), 5′-TCCTCCGCTTATTGATATGC). The primers were modified with the addition of “tags”, which are individual nucleotide sequences attached to both forward and reverse primers at their 5′ ends to create unique primer combinations, which can amplify the same ITS2 region but be distinguished from each other if PCR products produced by these primers are mixed together and subjected to high throughput sequencing; tags allow the results to be obtained for specific PCR products from the overall mix (multiplex) during the bioinformatic processing. Altogether 20 tagged primers were synthesized and utilized (Table 1). Each DNA sample in the amount ca 100 ng from each individual swab was subject to a PCR with a pair of primers carrying an individual tag. A negative control (water used in all reactions) and positive control, such as the SynMock collection, provided by Drs. J. Palmer and D. Lindner, USDA Forest services [57] were also amplified with specific tags. For the positive control, 9 mock fungal ITS sequences, with a total concentration of 0.1 ng per PCR reaction of the equimolar mix of SynMock plasmids [57], each at 0.1 ng/μL, were taken. Negative and positive controls were used to serve as controls for the contamination and specificity of the NGS reaction.

A PCR was performed using the PCR Supermix (ThermoFisher Scientific, MA, USA). The PCR conditions were 5 min at 94 °C, 30 cycles of 30 s at 94 °C, 30 s at 52 °C and 30 s 72 °C and the final 5 min at 72 °C. The success of PCR was confirmed by gel electrophoresis (see an example in Figure 2). The Mag-Bind RXN Pure Plus Quick kit (Omega Biotek, GA, USA) was used to purify the PCR products according to the manufacturer’s manual. The concentration of the purified tagged amplicons was measured using a Qubit 3.0 fluorimeter using ds DNA HS Assay kit (ThermoFisher Scientific, MA, USA). All PCR results were verified by agarose gel electrophoresis. All negative and positive controls were processed using the same methods as all other PCR products. In most cases, negative controls did not produce any visible bands; however, sometimes, slightly visible bands were detected (Figure 2). This was normal, considering the natural presence of molds in the non-sterile environment. Regardless of the detection of a band in the negative control of a gel, the whole volume of the finished negative control PCR was subjected to further processing with high throughput sequencing and bioinformatics analysis.

### 2.4. High-Throughput Sequencing (Next Generation Sequencing, NGS)

Purified tagged PCR amplicons were mixed together in a multiplex in equal 100 ng amounts; then, the sample was concentrated using Amicon Ultra-0.5 30 K centrifugal filters (Millipore Sigma, USA) for the volume and concentration required by the sequencing facility. DNA library preparations, sequencing reactions and adapter sequence trimming were conducted at Genewiz (now Azenta, South Plainfield, NJ, USA). DNA Library Preparation was performed using the NEBNext Ultra DNA Library Prep kit following the manufacturer’s recommendations (Illumina, San Diego, CA, USA). Briefly, end-repaired adapters were ligated after the adenylation of the 3′ends, followed by the enrichment of a limited cycle PCR. DNA libraries were validated and quantified before loading. The pooled DNA libraries were loaded on the Illumina instrument according to the manufacturer’s instructions. The samples were sequenced using a 2 × 250 paired-end (PE) configuration. Image analysis and base calling were conducted by the Illumina Control Software on the Illumina instrument.

### 2.5. Data Analysis

Our bioinformatics procedure was performed using the SCATA (Sequence Clustering and Analysis of Tagged Amplicons) pipeline (https://scata.mykopat.slu.se/) and using default settings as described in [56]. This pipeline is publicly available and is adapted to fungal community analysis based on ITS sequencing. In short, two FASTQ files were uploaded to SCATA, merged, unmatched sequences were removed and initial quality filtering was performed to remove sequences that were incomplete (i.e., miss one or both primers) or of low quality. The default settings of SCATA for quality trimming were the following: the minimum sequence length-to-keep after primer and tag trimming was 200, the minimum mean quality of bases in a read to keep was 20, and the minimum allowed base quality was 10. The sequences were then de-multiplexed based on the tag sequences to recover the sample identities. Once this was completed, amplicons were grouped by sequence similarity (clustered) into OTUs. For the ITS7–ITS4 primer combination, clustering was based on 41 bp of the conserved 5.8S region, about 105–330 bp of the ITS2 region and 38 bp of the large subunit ribosomal rRNA (LSU) gene. The number of sequences in each OTU represented the relative abundance of that OTU. The sequence similarity to the reference database was established using USEARCH (http://www.drive5.com/usearch/) as a search engine, and sequences were assembled into OTUs by single-linkage clustering. Parameters for clustering included a clustering distance of 0.015, a minimum alignment to consider clustering at 0.85, a mismatch penalty of 1, a gap open penalty of 0, a gap extension penalty of 1, an end gap weight of 0.0, collapse homopolymers 3, and a BLAST e-value cut off of 1 × 10^−60^. The UNITE 2019 database (http://unite.ut.ee) [57] was included in the clustering procedure as a reference database, plus an additional reference file containing 11 sequences SynMock.

Sequence data resolved to the species level were used for subsequent statistical analyses in R [58]. All analyses were performed on a filtered version of the dataset removing species found in the negative control, or that were not found in more than ten percent of the samples. The dataset was then checked for multivariate normality using Mardia’s test in the MVN package [59] prior to omnibus analyses. A non-parametric multivariate analysis of variance (PERMANOVA) was then used to determine if the fungal community composition differed significantly either based on the specimen ID (pigs 1–4), the location of the swab (mouth, abdomen, or tail), or the stage during which the swab was taken. Following omnibus testing, non-metric multidimensional scaling (NMDS) was used to evaluate differences between the treatments and time points. Indicator values were calculated and used to contrast species that contributed significantly to differences between stages using the labdsv package [60]. PERMANOVA and NMDS were performed using the vegan package [61]. To compare annotations for fungal species ecology, we retrieved available ecological roles and guild data for each species and its corresponding genus from the FUNguild database [62] using the fungarium package [63].

The full dataset of identified species with the corresponding metadata was shared via Zenodo’s open repository (www.zenodo.org) https://doi.org/10.5281/zenodo.8145408 [64].

## 3. Results

DNA was isolated from all collected user-assigned decay stages (Figure 1) and subjected to a PCR with the ITS7-ITS4 tagged primers. These primers resulted in an amplification of the ITS2 region of fungi. All PCR amplification results were verified using gel electrophoresis (Figure 2). SynMock was used as a positive control comprising a set of 11 plasmids that contained an artificially synthesized ITS2 fungal region to represent a synthetic community. This control served as a positive signal for both successful PCR reaction settings (Figure 2, lane 4) and NGS runs.

Relative abundance data for all species identified using SCATA were filtered to remove species that occurred in the negative controls or were exceedingly rare (occurring at low abundance in only one observed sample). This process reduced the number of fungal taxa included in the subsequent analysis from an initial 212 taxa to 180 taxa, 18% of which were resolved to the genus level, with the remaining 82% identified to the species level. These data were evaluated using Mardia’s test for multivariate normality and were found to not follow a normal distribution (*p* > 0.05), requiring the use of non-parametric omnibus tests in subsequent analyses. The species diversity at each stage of decay is presented in Figure 3.

We observed an overall change in the fungal community composition within our study, although no significant effect of the swabbing location was observed (*p* > 0.05, Figure 4), suggesting a similarity in fungal taxa contributing to decomposition across the body. Fungal community composition was observed to be significantly different between individual animals (F_(1,66)_ = 4.62, R^2^ = 0.055 *p* < 0.001), suggesting that each pig cultured a significantly different fungal community along its timeline of decomposition (Figure 5). More notably, however, when considering all pigs, the fungal community showed an interesting and significant difference across stages (F_(1,66)_ = 2.86, R^2^ = 0.172 *p* < 0.001). Evaluating the results of our NMDS, there appears to be a successional gradient within the fungal community (Figure 6), which results in the fungi observed in a given stage being more likely to be in the previous or following stage rather than in more distant stages (e.g., fungi observed in the bloat stage would be more likely to be observed in the fresh or decay stages but less likely to be observed in the skeletal stage). Together these results suggest that, while individuals might potentially contribute to changes in the fungal communities being observed, the decomposition stage has the strongest overall effect on shaping these communities. A total of 35 species were determined to significantly contribute to differences between each stage and served as indicator species for those stages (Table 2). The full dataset is shared via Zenodo’s open repository [64].

## 4. Discussion

Our study produced a dataset containing a list of fungal species participating in the decay of pig carcasses during the summer season along the East Coast of the US [64]. We determined that the fungal communities found on a dead body could vary widely based on the stage of decomposition the corpse was in, though broad patterns emerged when considering taxa that were active in a particular stage of decomposition; moreover, we observed distinct fungal taxa indicators for specific decomposition stages. There is an overall trend for increased fungal richness across the stages of decomposition (Figure 3), with lower fungal species richness when pigs are initially placed in the fresh stage. There is a significant increase in richness as decomposition progresses successively through bloating, decay and active decay stages, with significant changes in fungal taxa contributing to the different stages (Table 2), indicating important changes to the microbial fungal community as decomposition progresses. We believe that increased availability of the substrate during these stages, which is utilized by fungi, is one of the reasons explaining this diversity. Patterns observed in our multivariate analyses indicate that the fungal community changes in a successional pattern (Figure 6) with each stage being influenced by the preceding stage.

In addition to the overall appearance of unique and numerous fungal taxa with decomposition progression, we observed increased numbers of the indicator species in the later stages of decomposition as well (Table 2). This is in contrast to observations in other studies, such as Fu et al. [54], who compared fungal communities during the indoor and outdoor decomposition of pigs and found that, in outdoor carcasses, fungal diversity decreased as decomposition progressed. Interestingly, we did not record all the indicator fungal species that were dominant in Fu et al. [54] (*T. aurantiacus, C. xylopsoci,* and *Y. lipolytica*). This highlights that changes in biogeography likely contribute to the patterns involved in these processes, and such factors should be considered carefully when interpreting forensically relevant fungal species. In both Fu et al. and our studies, unspecified fungi from phylum Ascomycota were observed. Ascomycota is often present in association with animal decomposition [65]; in our case, we detected unresolved species from Ascomycota phylum only in the control and zero stages. Another species that we observed during the “zero” stage was *Alternaria metachromatica*; this fungus can grow on skin and mucous membranes and is quite common. The presence of *Alternaria* has even been recorded for mummies, indicating that this genus may be present naturally in pig development [66,67].

Overall, we found support for our hypothesis that the fungal community colonizing the decomposing pigs changes over time. Our second hypothesis that there would be no detectable difference between the body region sampled was also confirmed; we relate this fact to the colonization of carcasses by insects, which move vigorously across the decaying body distributing fungal material across the entire body. Only a weakly significant difference was detected between different pigs. Altogether, this implies that, while the stage of decomposition strongly influences the fungal community in a decomposing vertebrate, sampling methods do not need to be overly cautious in their design to account for such differences. Our discovery of no detectable difference between the body region sampled and only a weakly significant difference between individual animals could support the use of fungal succession analysis for the estimation of mPMI. It is worth noting, however, that regional or even seasonal differences might change this and promote differences not implied by this dataset.

Using this information, it might be possible to build a database of species that can assist forensic scientists and investigators in estimating mPMI when decomposing remains are found. By combining taxa observed to be active at particular stages of decomposition across spatial scales, this suggested database might make more detailed forensic interpretations possible.

While this work presents unique data for fungal taxa collected from decomposing specimens, much more work is required to fully evaluate the impacts of spatiotemporal factors in driving these changes. A uniform relationship between decomposing and fungal community diversity has yet to be demonstrated [55]. Future studies should include (1) the examination of the correlation between the advancement of fungi, bacteria and insects during the decomposition process; (2) the examination of correlations between fungal communities in surrounding soils or leaf matter or other environments with the fungal communities on the corpse, which could be helpful in the evaluation of the postmortem movement of a body; such studies would correlate with fungal biodiversity assessments, e.g., in our case for New Jersey, USA [68]; (3) evaluating the process of microbial decomposition in different geographical locations.

By viewing the decomposition by fungi through this type of larger lens, we can more discretely understand the combination of drivers promoting decomposition and continue to expand to fungal communities of human cadavers [44,69]. We also suggest that the method used here, employing widely available molecular biology resources and convenient analysis platforms, is a good template for a standard approach as the use of fungal taxa in forensic science moves forward.

## Figures and Tables

**Figure 1 jof-09-00866-f001:**
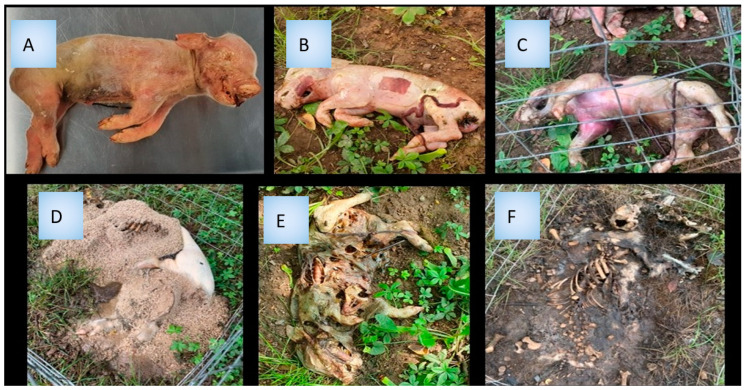
Six stages of swine carcass decomposition. (**A**) Appearance before placement, (**B**) Fresh, (**C**) Bloat, (**D**) Active decay, (**E**) Decay, (**F**) Skeletal.

**Figure 2 jof-09-00866-f002:**
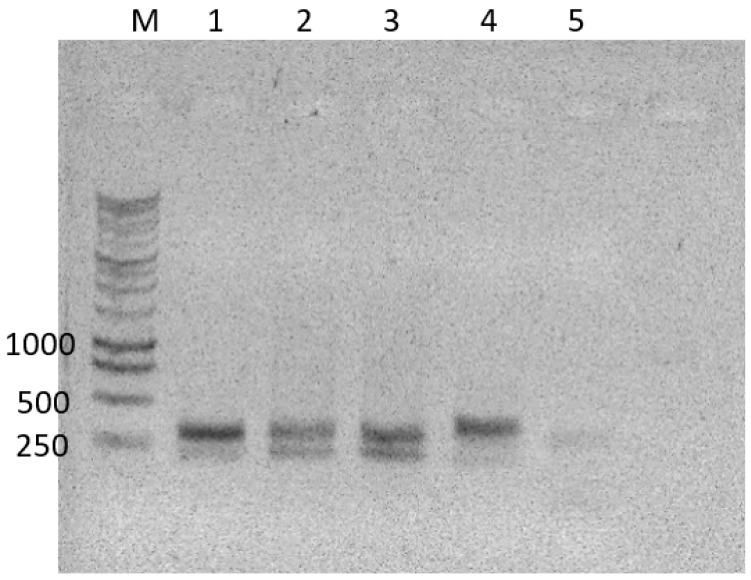
ITS2 fragment amplified from DNA extracted from the skeletal stage of pig 3, 1: head, primers tag_18F/R, 2: abdomen, primers tag_20F/R, 3: genitals, primers tag_21F/R, 4: positive control with SynMock, primers tag_22F/R, 5: negative control (water), primers tag_23F/R. M: 1 kb DNA ladder (bp, Promega). This gel is representative of all other amplification experiments conducted.

**Figure 3 jof-09-00866-f003:**
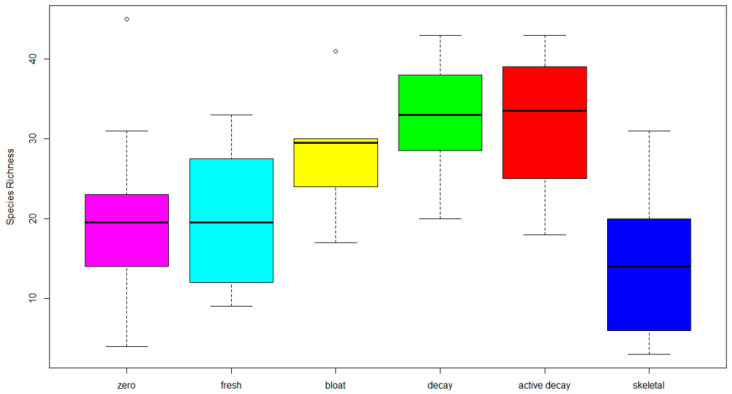
Fungal species richness for each stage of decomposition. Box plots: middle line, median; box, interquartile range; whiskers, 5th and 95th percentiles.

**Figure 4 jof-09-00866-f004:**
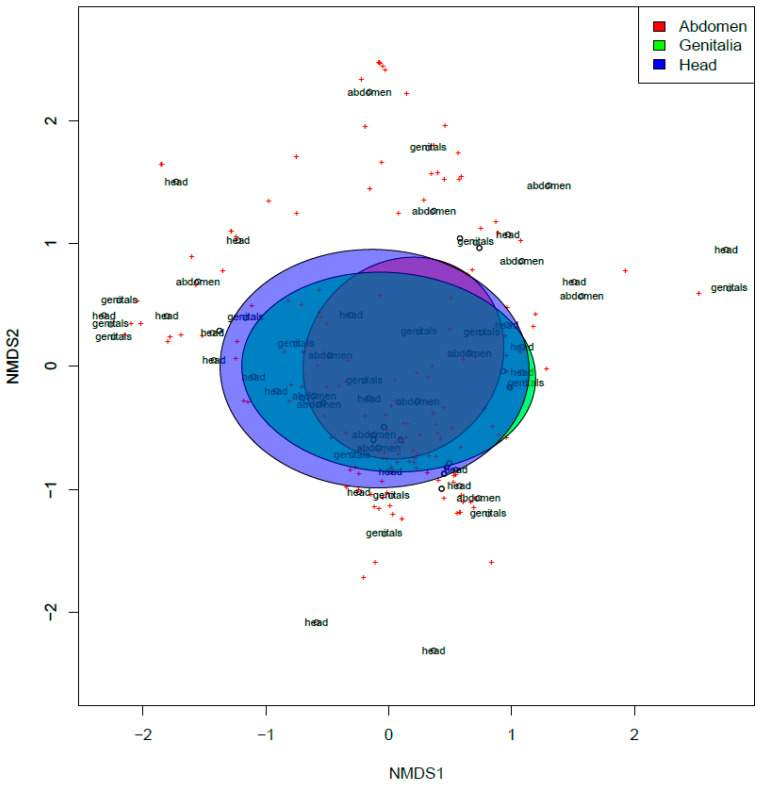
Non-metric multidimensional scaling (NMDS) plot for total fungal communities observed from the three locations assessed in the decomposition assay. The ellipses represent 95% confidence intervals. Assessed locations: red, abdomen; blue, head; green, genitalia.

**Figure 5 jof-09-00866-f005:**
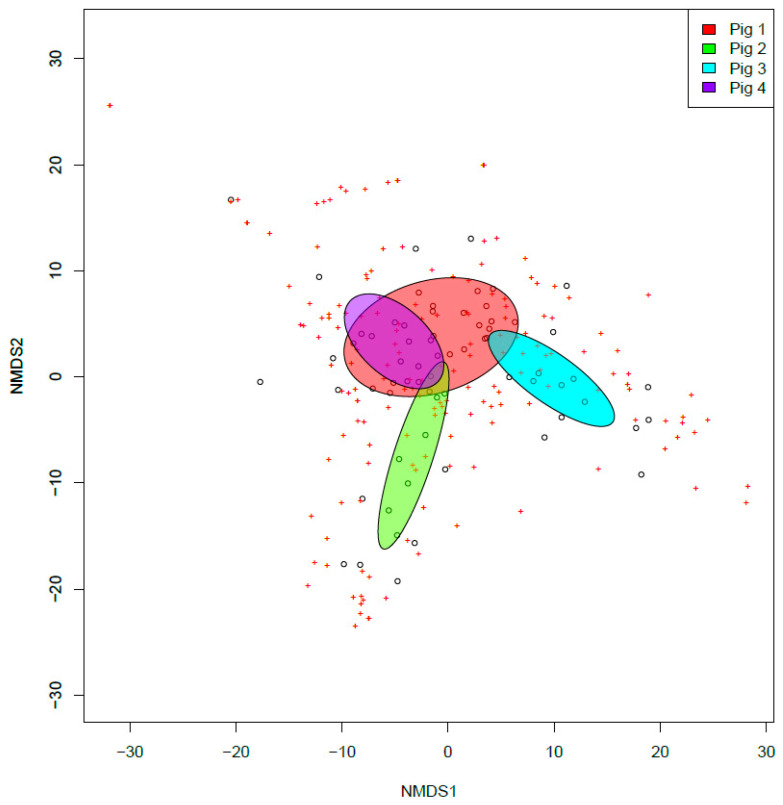
Non-metric multidimensional scaling (NMDS) plot for total fungal communities observed for each of the four pigs included in the decomposition assay. Ellipses represent 95% confidence intervals. Red, pig 1; green, pig 2; teal, pig 3; blue, pig 4.

**Figure 6 jof-09-00866-f006:**
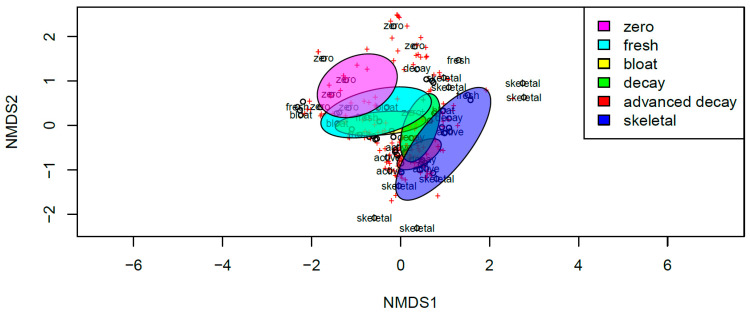
Non-metric multidimensional scaling (NMDS) plot for total fungal communities observed within the 6 user defined stages (zero, fresh, bloat, decay, active decay, skeletal) included in the assay. Ellipses represent 95% confidence intervals. Pink, stage zero; teal, stage fresh; yellow, stage bloat; green, stage decay; red, stage advanced decay; blue, stage skeletal.

**Table 1 jof-09-00866-t001:** Sequences of the primers used in the experiment.

Primer Name	Primer Sequence, from 5′
tag_1F	CACACGATCT GTGARTCATCGAATCTTTG
tag_1R	CACACGCTGT TCCTCCGCTTATTGATATGC
tag_4F	CACATAGTCT GTGARTCATCGAATCTTTG
tag_4R	CACATGTCGT TCCTCCGCTTATTGATATGC
tag_5F	CACATGACTT GTGARTCATCGAATCTTTG
tag_5R	CACGCAGCAT TCCTCCGCTTATTGATATGC
tag_6F	CACGATCAGT GTGARTCATCGAATCTTTG
tag_6R	CACTAGCGCT TCCTCCGCTTATTGATATGC
tag_7F	CACGTGCTCT GTGARTCATCGAATCTTTG
tag_7R	CACTATGCAT TCCTCCGCTTATTGATATGC
tag_8F	CACTATAGCT GTGARTCATCGAATCTTTG
tag_8R	CACTCACACT TCCTCCGCTTATTGATATGC
tag_9F	CACTATGTGT GTGARTCATCGAATCTTTG
tag_9R	CACTCTGAGT TCCTCCGCTTATTGATATGC
tag_10F	CACTCAGAGT GTGARTCATCGAATCTTTG
tag_10R	CACTGATCAT TCCTCCGCTTATTGATATGC
tag_11F	CACTCTCACT GTGARTCATCGAATCTTTG
tag_11R	CACTGTATGT TCCTCCGCTTATTGATATGC
tag_12F	CACTGCTACT GTGARTCATCGAATCTTTG
tag_12R	CAGACATAGT TCCTCCGCTTATTGATATGC
tag_13F	CAGACAGTGT GTGARTCATCGAATCTTTG
tag_13R	CAGACTATGT TCCTCCGCTTATTGATATGC
tag_14F	CAGACATCTT GTGARTCATCGAATCTTTG
tag_14R	CAGAGCTCAT TCCTCCGCTTATTGATATGC
tag_15F	CAGAGACGCT GTGARTCATCGAATCTTTG
tag_15R	CAGATACACT TCCTCCGCTTATTGATATGC
tag_16F	CAGAGCTCGT GTGARTCATCGAATCTTTG
tag_16R	CAGATGCTAT TCCTCCGCTTATTGATATGC
tag_17F	CAGAGTATGT GTGARTCATCGAATCTTTG
tag_17R	CAGCACGACT TCCTCCGCTTATTGATATGC
tag_18F	CAGATACAGT GTGARTCATCGAATCTTTG
tag_18R	CAGCACTCGT TCCTCCGCTTATTGATATGC
tag_20F	CAGCACTATT GTGARTCATCGAATCTTTG
tag_20R	CAGCTGACGT TCCTCCGCTTATTGATATGC
tag_21F	CAGCGATACT GTGARTCATCGAATCTTTG
tag_21R	CAGTATCTCT TCCTCCGCTTATTGATATGC
tag_22F	CAGCTAGATT GTGARTCATCGAATCTTTG
tag_22R	CAGTCAGTCT TCCTCCGCTTATTGATATGC
tag_23F	CAGCTCACTT GTGARTCATCGAATCTTTG
tag_23R	CAGTCGTGCT TCCTCCGCTTATTGATATGC

**Table 2 jof-09-00866-t002:** A summary table for species determined to have significant indicator values for each of the stages across our decomposition timeline.

Fungal Species	Stage	Indicator Value	*p*-Value	Guild
*Alternaria metachromatica*	Zero	0.450013	0.029	
*Apiotrichum mycotoxinovorans*	Zero	0.3204	0.005	
*Ascomycota* sp.	Zero	0.30861	0.004	
*Beauveria bassiana*	Zero	0.381168	0.044	Animal pathogen
*Cladosporium tenuissimum*	Zero	0.279405	0.029	Endophyte
*Cryptococcus uniguttulatus*	Zero	0.3	0.037	
*Cystobasidium pallidum*	Zero	0.616563	0.016	
*Cystofilobasidium capitatum*	Fresh	0.24132	0.002	
*Erysiphe corylacearum*	Fresh	0.558174	0.004	
*Exophiala salmonis*	Bloat	0.223837	0.021	Endophyte
*Filobasidium* sp.	Bloat	0.173077	0.032	
*Gibellulopsis piscis*	Bloat	0.370083	0.038	
*Jaminaea* sp.	Bloat	0.404757	0.003	
*Knufia tsunedae*	Bloat	0.329308	0.028	Soil Saprotroph
*Lachancea thermotolerans*	Bloat	0.325791	0.003	
*Leucosporidium intermedium*	Bloat	0.278229	0.016	
*Malassezia restricta*	Bloat	0.341931	0.035	Animal pathogen
*Metarhizium anisopliae*	Bloat	0.229167	0.003	
*Mortierella alpina*	Decay	0.352564	0.001	
*Mortierella elongata*	Decay	0.526913	0.016	
*Mucor racemosus*	Decay	0.25907	0.006	Endophyte
*Neophaeococcomyces catenatus*	Decay	0.264503	0.043	
*Phallus rugulosus*	Decay	0.214286	0.001	
*Pleosporales* sp.	Decay	0.25	0.011	
*Powellomyces* sp.	Decay	0.351021	0.004	Plant pathogen
*Pseudomicrostroma phylloplanum*	Decay	0.361429	0.03	
*Pyrenochaetopsis leptospora*	Decay	0.242138	0.001	
*Schwanniomyces occidentalis*	Decay	0.734502	0.019	
*Tausonia pullulans*	Decay	0.296296	0.003	Saprotroph
*Trichoderma harzianum*	Decay	0.480422	0.036	Endophyte-Fungal Parasite-Plant Pathogen
*Trichomeriaceae* sp.	Decay	0.333857	0.015	Epiphyte
*Ustilago nunavutica*	Decay	0.457515	0.001	
*Wallemia muriae*	Decay	0.257678	0.001	
*Wallemia tropicalis*	Decay	0.654491	0.002	
*Xylariales* sp.	Active decay	0.267912	0.027	

## Data Availability

The full dataset of identified species with the corresponding metadata was shared via Zenodo’s open repository (www.zenodo.org) https://doi.org/10.5281/zenodo.8145408.

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
