# Peer review of "Assessment of Fungal Succession in Decomposing Swine Carcasses (Sus scrofa L.) Using DNA Metabarcoding"

_jof, 2023, doi:10.3390/jof9090866_

Round 1

Reviewer 1 Report

Review of “Assessment of fungal succession in decomposing swine carcasses (Sus scrofa L.) using DNA metabarcoding” by Gemmellaro et al.

The manuscript by Gemmellaro et al. investigates the fungal community composition at various stages of decomposition on pigs in the Eastern USA. Using metabarcoding of the fungal ITS2 region they explore the successional patterns of fungal communities across the decomposition process from fresh carcass through the skeletal phase. Their study aims examine the successional patterns of fungi during the decomposition of swine carcasses in the Northeastern United States. This study will explore the biodiversity of fungi, changes in fungal taxa during decomposition, and the distribution of fungi throughout the body during decomposition. In summary, they aim to examine the decomposition process, with a particular focus on the role of fungi to gain a better understanding of decomposition to aid forensic investigations, especially in determining the time of death.

Gemmellaro and colleagues observed an overall change in the fungal community composition across the decomposition stages, with each pig fostering a distinct fungal community. While individual variation was noted, the decomposition stage was found to have the strongest effect on shaping these communities. A total of 35 species were identified as significantly contributing to differences between each stage, serving as indicator species for those stages.

78: Species of insect or species being decomposed?

120: You use the UNITE database in your analysis. Why not mention it here instead of or along with GenBank.

121-122: How does this method provide “complex microbiome” information if you are amplifying the barcode sequence?

155-158: Bodies were checked twice daily. What was being checked? Were samples taken twice a day or just at the beginning of each stage?

161: Why were only 4 of the 12 pigs used in the study?

163: How many total samples were processed?

174-175: Please consider rephrasing this sentence. I understand what you did, but the way it is worded is a little confusing.

178 and throughout: “Error! Reference source not found.”

177: It indicated that 10 tagged primer pairs are used, but there are 20 primer pairs listed in Table 1. What are the other 10 primer pairs used for?

210: What quality score was used as a cutoff?

212: At what similarity value were sequences clustered? What method was used to cluster sequences? Did you used a reference database to cluster or were they clustered de novo?

212-214: Why include the conserved regions in your clusters? Why not remove the conserved 5.8S and LSU regions before clustering? ITSx is an easy-to-use open-source software to extracting just the variable ITS sequences.

283-284: Were indicator species found in all individual pigs are each of the phases?

Overall, the manuscript is well written and clear. The topic is interesting and has the potential for use in forensics but may have some limitation as location may influence which fungi are present. Additionally, there are a lot of instances were it seems like there should be a citation but it says “Error! Reference source not found.”  

English is great. There are few very minor errors here and there, but I didn't spend time editing. 

Author Response

Dear reviewer,

We would like to thank you for critical reading of the manuscript. Your comments are very important and we hope while addressing them we were able to improve the manuscript. We sincerely hope that now it is acceptable for publication.

Our response:

78: Species of insect or species being decomposed?

Answer: insect species, the text was corrected.

120: You use the UNITE database in your analysis. Why not mention it here instead of or along with GenBank.

Answer: we have added UNITE and a reference to it as well here.

121-122: How does this method provide “complex microbiome” information if you are amplifying the barcode sequence?

Answer: Metabarcoding is a method that is especially useful when analyzing microbial communities, such as bacteria and fungi; very often there is no other method to assess microbiome diversity. Bacterial or fungal species might be visualized for identification by morphology after culturing, but there is no guarantee that all species from a community would grow in culture since the artificial culturing conditions often do not exactly represent the native substrate and conditions. Therefore one of the meat ways to assess more complex distributions of microbes is employing methods like ours. Reference 49, Taberlet et al, is a book on metabarcoding of many species and its use in the analysis of environmental communities. We have added more references so the readers can familiarize themselves with this method if they are less familiar with its applications.

155-158: Bodies were checked twice daily. What was being checked? Were samples taken twice a day or just at the beginning of each stage?

Answer: Bodies were checked twice a day to assess the decomposition stage and observe insect colonization patterns. Fungal samples were collected at the beginning of stage using the methods outlined in the text. Depending on the weather, the duration of each stage could last a different time, hence we had to check the bodies often.

161: Why were only 4 of the 12 pigs used in the study?

Answer: We used 4 pigs to further randomize the experiment. Variation in weather patterns, or even more surprising events (sometimes the pig is stolen by a vulture even if it is protected by a cage, etc) mean that the experiences of the decomposing specimens can vary. We selected 4 random pigs from the total number used in the larger experiment and observed the same pattern in all four, considering this number of samples to be sufficient for our experiment. 

163: How many total samples were processed?

Answer: 75 samples in total.

174-175: Please consider rephrasing this sentence. I understand what you did, but the way it is worded is a little confusing.

Answer: the sentence was revised for clarity.

178 and throughout: “Error! Reference source not found.”

Answer: This error is caused by the formatting in the MDPI system. While preparing the manuscript, we used a system of cross-references embedded in Word software to organize figures and tables. The captures to the figures contained a hyperlink to the figure number, and in the text every time we refer to the figure we insert a cross link to this figure number. The file uploaded to the MDPI system contained the cross-references which were not recognized by the MDPI editing system which converted the manuscript file into MDPI preferred format, hence the hyperlinks became inactive and the file gave an error. We apologize for inconvenience and will remove all embedded references from the revised manuscript.

177: It indicated that 10 tagged primer pairs are used, but there are 20 primer pairs listed in Table 1. What are the other 10 primer pairs used for?

Answer: We apologize for the typo, indeed 20 primer pairs.

210: What quality score was used as a cutoff?

Answer: The analysis of sequences was performed in SCATA pipeline, which is an online pipeline designed to work with multiplexes of tagged ITS sequences specifically. We used default settings for all analyses (honestly we tried to play with the settings and not much changed in the output). We have added a couple of sentences to better describe SCATA. More on SCATA can be found here, reference 55:

Clemmensen, K.E.; Ihrmark, K.; Durling, M.B.; Lindahl, B.D. Sample Preparation for Fungal Community Analysis by High-Throughput Sequencing of Barcode Amplicons. Methods in molecular biology (Clifton, N.J.) 2016, 1399, 61-88, doi:10.1007/978-1-4939-3369-3_4.

The default settings of SCATA for the initial sequence quality trimming are the following: the minimum sequence length to keep after primer and tag trimming was 200, minimum mean quality of bases in a read to keep was 20, minimum allowed base quality was 10. 

212: At what similarity value were sequences clustered? What method was used to cluster sequences? Did you used a reference database to cluster or were they clustered de novo?

Answer: Again, we used default settings of SCATA. Threshold distance for where clustering should occur was at 0.015, the minimum length of pairwise alignment in the clustering process required to consider a sequence pair for clustering was 0.85. USEARCH with single-linkage clustering was used to assemble sequences in OTU. We used the UNITE 2019 database as a reference database. This information is present in the manuscript in the same paragraph so readers can interpret our results based on the settings used.

212-214: Why include the conserved regions in your clusters? Why not remove the conserved 5.8S and LSU regions before clustering? ITSx is an easy-to-use open-source software to extracting just the variable ITS sequences.

Answer: this is because primers fITS7 and ITS4 produce ITS2 sequence with the 5.8S and LSU “overhangs”. This clustering procedure is performed by SCATA, pretty much an automated pipeline, and this parameter we cannot change. SCATA greatly simplifies the work with NGS files that contain multiple PCR products with tags, hence it is our pipeline of choice. 

283-284: Were indicator species found in all individual pigs are each of the phases?

Answer: The Indicator values calculate across replicates - essentially highlighting taxa that are noteworthy in the context of the experimental design. There are not always indicator taxa for every stage but when the indicator analysis approach highlights a taxon it is usually because they are observed in all replicates for that stage/timepoint, as is the case for our data. As with any statistical analysis - there are sometimes replicates that might not follow a trend for a particular dependent variable, but the analysis used here help highlight useful trends for interpretation.

Reviewer 2 Report

Authors investigated about the fungal species involved in the different stage of decomposition of swine carcasses trying to understand the combination of drivers promoting the process also in human cadavers.

In my opinion the work is suitable for publication even if I would have some questions for authors:

1. Can you please specify the age of juvenile pigs? How long after birth did they die?

2. Were pigs breastfed or artificially fed?

3. How long after pigs death was time zero taken? 

4. How they were stored after death?

Please specify all these information in Material and Methods section.

Moreover, please fix the following errors I have found in the text:

Line 41: please check the font of the sentence

Lines 178 and 187, 245 and 248, 265: please check references

Line 252: please improve the quality of Fig. 2. A band is visible in the negative control, I suggest to choose another image of gel electrophoresis

Lines 309-320: check the font and references

Author Response

Dear reviewer,

We would like to thank you for critical reading of the manuscript. Your comments are very important and we hope while addressing them we were able to improve the manuscript. We sincerely hope that now it is acceptable for publication.

Our response:

  1. 1. Can you please specify the age of juvenile pigs? How long after birth did they die?; 2. Were pigs breastfed or artificially fed?

Answer: Pigs were obtained from UPenn swine research facility; we do not have information on their diet, on time of death or on their exact age at death. We have added the facility to the Methods.

  1. 3. How long after pigs death was time zero taken? 

Answer: time zero was recorded after the carcasses were taken out of the freezer and when they were completely thawed (from 4 to 6.5h)

  1. How they were stored after death?

Answer: At -20℃ , we have added this information to the text. 

Moreover, please fix the following errors I have found in the text:

Line 41: please check the font of the sentence

Answer: fixed

Lines 178 and 187, 245 and 248, 265: please check references

Answer: This error is caused by the formatting in the MDPI system. While preparing the manuscript, we used a system of cross-references embedded in Word software to organize figures and tables. The captures to the figures contained a hyperlink to the figure number, and in the text every time we refer to the figure we insert a cross link to this figure number. The file uploaded to the MDPI system contained the cross-references which were not recognized by the MDPI editing system which converted the manuscript file into MDPI preferred format, hence the hyperlinks became inactive and the file gave an error. We apologize for inconvenience and will remove all embedded references from the revised manuscript.

Line 252: please improve the quality of Fig. 2. A band is visible in the negative control, I suggest to choose another image of gel electrophoresis

Answer: This is a valuable comment and we want to assure the reviewer that we thoroughly considered it but would prefer to maintain the existing graphic, if possible. While no detectable band in negative control  is a desirable output of any PCR, it is not always possible to achieve it. We work in a non-sterile environment, and contamination with molds is always possible since the method is very sensitive. The way around it is to process the negative control as an individual PCR product (even if there is no band visible!) through the NGS and bioinformatics pipeline, and then remove any species identified in the negative control from all samples that were processed during the same day with the same reagents as the negative control. We would like to keep Figure 2 because it shows results as they appear during processing and highlights why we need to process through NGS negative controls as well. We processed them regardless of the visibility of the band (what if there is still contamination, but not much so we do not see it on the gel? But it will come up in NGS).

The following text explains the negative control handling in the manuscript:

In Materials and Methods, PCR:

All negative and positive controls were processed in the same way with all other PCR products. In most cases, negative controls did not produce any visible bands, however, sometimes slight visible bands were detected (Figure 2). This is normal considering the natural presence of molds in a non-sterile environment. Regardless of detection of a band in negative control in a gel, the whole volume of the finished negative control PCR was subjected to further processing with high throughput sequencing and bioinformatics analysis.

In Materials and Methods, Data analysis:

All analyses were performed on a filtered version of the dataset removing species found in the negative control or that were not found in more than ten percent of samples.

In Results:

Relative abundance data for all species identified using SCATA were filtered to remove species that occurred in the negative controls or were exceedingly rare (occurring at low abundance in only one observed sample). 

Lines 309-320: check the font and references

Answer: fixed